# Extraction of Pest Insect Characteristics Present in a Mirasol Pepper (*Capsicum annuum* L.) Crop by Digital Image Processing

**Mireya Moreno-Lucio** [1], **Celina Lizeth Castañeda-Miranda** [1], **Gustavo Espinoza-García** [2],
**Carlos Alberto Olvera-Olvera** [2], **Luis F. Luque-Vega** [3], **Antonio Del Rio-De Santiago** [1],
**Héctor A. Guerrero-Osuna** [1], **Ma. del Rosario Martínez-Blanco** [1] and **Luis Octavio Solís-Sánchez** [1,*]

[1] Posgrado en Ingeniería y Tecnología Aplicada, Unidad Académica de Ingeniería Eléctrica, Universidad Autónoma de Zacatecas "Francisco García Salinas", Zacatecas 98000, Mexico; 34156720@uaz.edu.mx (M.M.-L.); celina.castaneda@outlook.com (C.L.C.-M.); adelrio22@gmail.com (A.D.R.-D.S.); hectorguerreroo@uaz.edu.mx (H.A.G.-O.); mrosariomb@uaz.edu.mx (M.d.R.M.-B.)

[2] Unidad Académica de Ingeniería Eléctrica, Universidad Autónoma de Zacatecas "Francisco García Salinas", Zacatecas 98000, Mexico; gustavoesga@hotmail.com (G.E.-G.); colvera@uaz.edu.mx (C.A.O.-O.)

[3] Centro de Investigación, Innovación y Desarrollo Tecnológico CIIDETEC-UVM, Universidad del Valle de Mexico, Guadalajara 44100, Mexico; luis.luque@uvmnet.edu

\* Correspondence: lsolis@uaz.edu.mx

**Abstract:** One of the main problems in crops is the presence of pests. Traditionally, sticky yellow traps are used to detect pest insects, and they are then analyzed by a specialist to identify the pest insects present in the crop. To facilitate the identification, classification, and counting of these insects, it is possible to use digital image processing (DIP). This study aims to demonstrate that DIP is useful for extracting invariant characteristics of psyllids (*Bactericera cockerelli*), thrips (*Thrips tabaci*), whiteflies (*Bemisia tabaci*), potato flea beetles (*Epitrix cucumeris*), pepper weevils (*Anthonomus eugenii*), and aphids (*Myzus persicae*). The characteristics (e.g., area, eccentricity, and solidity) help classify insects. DIP includes a first stage that consists of improving the image by changing the levels of color intensity, applying morphological filters, and detecting objects of interest, and a second stage that consists of applying a transformation of invariant scales to extract characteristics of insects, independently of size or orientation. The results were compared with the data obtained from an entomologist, reaching up to 90% precision for the classification of these insects.

**Keywords:** insect features; mirasol pepper; digital image processing; pests

## 1. Introduction

Agriculture 4.0 replaces traditional production methods and global agricultural strategies by integrating them into a value chain [1], allowing the agronomic industry to seek the interconnection of the systems available for the field and adaptability of production systems, improving the rotation of crops to achieve a higher level of production and efficiency of production systems by optimizing the efficient use of water, fertilizers and phytosanitary products. All of this gives rise to precision agriculture, which aims to optimize the management of a plot from the agronomic, economic and environmental points of view based on observation, measurement and action in cultivation fields to provide higher yields in production with a lower cost in inputs, leading to a reduction in environmental pollution and labor [2,3].

In the environmental area of precision agriculture, the aim is to reduce the impact linked to agricultural activity [4] and to reduce the cost of labor. This area includes three main phases: optimization in the appropriate use of water, appropriate fertilizers and the appropriate use of phytosanitary products.

To carry out the proper use of phytosanitary products, it is necessary to clean out weeds, have knowledge of the type and density of pests, and, above all, know the types of

phytosanitary products that should be used to control different types of insects and the correct quantities to apply according to the density of the pests present. The intention of this is to reduce or eliminate the use of phytosanitary products and minimize the impact on the environment. The presence of pests in open-air crops and under greenhouse conditions is one of the biggest problems faced by producers at a national and international level since it can cause losses of up to 40% of the sown crop [5].

In recent years, the presence of different pest insects, such as psyllids (*Bactericera cockerelli*), thrips (*Thrips tabaci*), whiteflies (*Bemisia tabaci*), potato flea beetles (*Epitrix cucumeris*), pepper weevils (*Anthonomus eugenii*), and aphids (*Myzus persicae*), has been of greater importance for detection because these types of insects are vectors of viruses and phytoplasmas that affect the physiology and growth of mirasol pepper and interfere with the production of this crop [6–11].

In various regional studies, up to 55 species of insects associated with the cultivation of chili have been detected; however, only some are considered important pests. In this study, we describe some of the most important pests for mirasol pepper crops in México, according to Macías Valdez et al. (2010) [6]. The importance of these species is detailed below.

*Bactericera cockerelli*: This is widely distributed in regions of Solanaceae producers; its importance lies in the direct damage it causes when sucking the sap of plants. It is a vector of diseases, such as the permanent tomato, potato purple top, and zebra chip [12–14].

*Thrips tabacci*: This insect damages more than 500 types of fruit plants, vegetables and ornamental plants [15] and is found in most of the world. It is extremely dangerous for agriculture [16] since if it is not fought in time, it can cause crop losses of 70% to 100% of production, by either direct or indirect damage [17]. The insect feeds mainly on flowers and severely damages vegetative shoots, inflorescences and fruits in formation, limiting the appearance quality of the product by 25%. In addition, the insect is a vector of viruses such as Tospovirus (Bunyaviridae) and tomato spotted wild virus (TSWV) [16].

*Bemisia tabaci*: This species lays its eggs on the underside of the leaf. The larvae commonly feed on the phloem of the plant and suck the sap, damaging the leaf. They also affect the plant by transmitting viruses such as begomovirus, which causes yellowing of the leaves, defoliation and reduction in vigor and can cause crop losses of 20% to 100% [6,18]. In crops such as tomatoes, they transmit the tomato yellow leaf curl virus (TYLCV), which limits tomato production in various regions [19].

*Epitrix cucumeris*: This species damages a great diversity of crops, such as beets, squash, chili, tomato, and others. It feeds on the tender leaves and shoots, especially seedlings recently established in the field. During winter, it lives as an adult under leaves, grass, or undergrowth around crop fields. In spring, they leave their hiding places and begin to feed on the vegetation that starts the new cycle of the available crop [20,21].

*Anthonomus eugenii*: The larvae of this pest develop inside the fruit and feed inside the fruit, which causes a premature change in color and fruit fall. When the adults emerge from the fruit, they leave holes through which moisture penetrates, causing deformity and rot [6]. A loss of up to 70% in chili crops is associated with this insect [22].

*Myzus persicae*: This species is one of the most widely distributed pests in the world. It colonizes more than 30 botanical families and is present in crops such as citrus orchards [23] and bell pepper crops [24]. The damage is due to the suction of sap from the phloem and the transmission of up to 100 plant viruses [25].

In 1998, Zayas and Flinn described digital imaging techniques to identify insects in bulk wheat grain samples; they used multispectral analysis in combination with pattern recognition techniques, recognizing insects such as *Rhyzopertha dominica beetles* [26]. In 2008, Boissard et al. [7] described a cognitive vision approach for the early detection of pests in greenhouse crops. Their research described the use of digital processing of leaf images of the plants in which pests, such as the whitefly, are found, in order to be used in situ.

Artificial vision has proven effective in detecting different types of pests in crops through the use of sticky yellow traps. In 2009, Solis-Sánchez et al. [27] proposed computer vision techniques to detect whiteflies (*Bemisia tabaci*), using an algorithm described as

LOSS, in which they used characteristics such as the solidity, area, and eccentricity of the objects detected. In 2011 [28], the same author described a second version of his LOSS algorithm, named in this publication the LOSS V2 algorithm, which includes SIFT in addition to the characteristics already used by the algorithm in its initial version. Liu et al., in 2016, described a method of identification of aphids and monitoring of populations based on digital images. They described the use of descriptors, such as histograms and stable extremal region, oriented gradient features, and an aphid identification model [29]. Maharlooei, Sivarajan, Bajwa, Harmon, and Nowatzki, in 2017, described digital image processing techniques for the detection and counting of aphids present in leaves of soybean crops in a greenhouse; the images were taken under different lighting conditions and processed in MATLAB. Digital processing included grayscaling of images and alterations to the hue, saturation, and intensity [30].

This paper describes a digital image processing method for the extraction of invariant features of common insect pests that attack vegetable crops. The paper presents the specific descriptors utilized for each of the pest categories, which allowed us to detect, classify, and count insects in traps.

## 2. Materials and Methods

In practice, the most common detection procedure is the use of sticky yellow traps placed in the culture [27,28,31,32], which are visually explored with the help of microscopes or magnifying glasses. This work is done by entomological specialists with years of experience and knowledge in the diversity of insects that affect crops. For this process, minutes or even hours are invested in each trap since the pest insects are very small [33]. This, in conjunction with the characteristics of the air, which can dirty each trap with dust, increases the level of complexity when detecting trapped insects [34].

For the detection of the different types and incidence rates of insect pests, yellow sticky traps were placed 10 cm above the plant canopy of pepper plants grown under irrigation.

In order to know the density and type of pests trapped in the experiment, the collaboration of the specialist in charge of the entomology laboratory of the National Institute of Forestry, Agricultural and Livestock Research (INIFAP, Norte Centro, Zacatecas, México) was requested.

The specialist counted and classified the insects trapped in each trap throughout the sampling period.

Pests, such as psyllids (*Bactericera cockerelli*), thrips (*Thrips tabaci*), whiteflies (*Bemisia tabaci*), potato flea beetles (*Epitrix cucumeris*), pepper weevils (*Anthonomus eugenii*), and aphids (*Myzus persicae*), were detected. Each trap was tagged and sorted according to the date of collection. Digital image processing was developed on the MATLAB platform (The Mathworks, Inc., Natick, MA, USA) to extract the invariant features of insects, using database images taken from the sticky traps located in the crop area. The experiment began with the distribution of 20 yellow sticky traps placed in the plot; the traps were collected at the end of 7 days, for 19 consecutive weeks.

The digital image processing consisted of two stages, which are explained below.

### 2.1. First Stage: Scale-Invariant Feature Transform (SIFT)

Scale-invariant feature transform (SIFT) is a method that was described by David G. Lowe in 1999 [35]. It is used to extract features invariant in orientation and scale in images.

This approach transforms an image into a large number of local feature vectors, each of which is invariant to translation, scale or size, illumination, and image rotation. The feature vectors are then used to detect objects and to obtain other descriptors, such as signature or chain, which were described by Moreno-Lucio et al. in 2019 and Solís-Sánchez et al. in 2011 [28,36].

The SIFT algorithm consists of five stages (Figure 1), described and explained by P. Flores and J. Braun in 2011 and D. G. Lowe (1999, 2004) [9,10,13,14,37–39]. These five stages are applied to an original image and to another image that has the same characteristics.

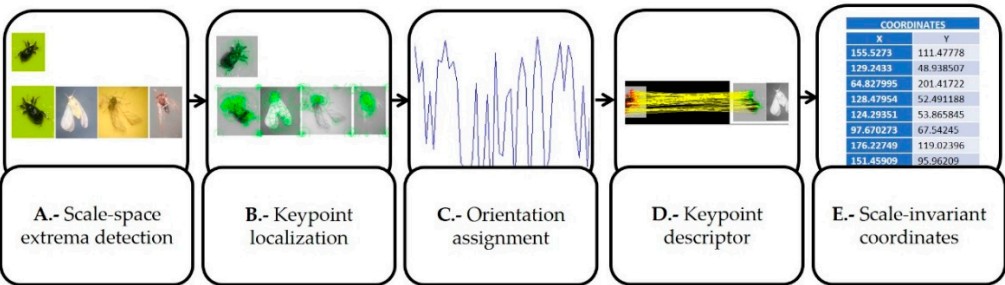

**Figure 1.** Description of the steps of the SIFT algorithm.

We used the SIFT algorithm to examine the features of the insects in different positions, sizes and orientations in the sticky cards. This procedure provided the information that was later used to detect objects and produce specific descriptors.

The stages of the SIFT algorithm are described below.

### 2.1.1. Scale-Space Extrema Detection

We implemented a difference-of-Gaussian function to identify points of interest with values invariant in scale and orientation (Figure 1A).

### 2.1.2. Key Point Localization

This identifies locations in image-scale space that are invariant points of scale. The image translation, scaling, rotation, noise, or distortion is obtained (Figure 1B).

### 2.1.3. Orientation Assignment

The location of one or more key points is assigned. This is based on the directions of the image gradient providing invariance in transformations in relation to orientation, scale, and location assigned for each feature (Figure 1C).

### 2.1.4. Key Point Descriptor

The gradients of the local image are measured on the selected scale in the region of each key point (Figure 1D).

### 2.1.5. Scale-Invariant Coordinates

In the last step, a descriptor of $4 \times 4 \times 8 = 128$ values is obtained for each key point, the positioning of which is given by coordinates found within the image (Figure 1E).

### 2.2. Second Stage: Color Transformation and Regions of Interest

In the second stage of image processing, the correction and extraction of specific characteristics of each of the insects are carried out. Characteristics of each insect that differentiate one genus from other are extracted, and the techniques and filters applied are as follows.

### 2.2.1. Grayscale

Equation (1) is applied to the original image to separate it by color layers. To convert it into grayscale, in which the information matrix and saturation are eliminated while retaining the luminance, a weighted sum of the R, G, and B components of the image is formed according to Equation (2).

$$R = image \ (:, :, 1);$$
$$G = image \ (:, :, 2); \qquad (1)$$
$$B = image \ (:, :, 3);$$

$$Grayscale = 0.2989 * R + 0.5870 * G + 0.1149 * B \qquad (2)$$

### 2.2.2. CIE*L*A*B

The goal of transforming the gray image to the CIELAB color space (Equation (3)) is to produce a color-changing image with similar visual significance as another color space. *L* is the brightness from black to white, A goes from red to green, and B is a gradient of the color blue [40,41].

$$CIE\ LAB = 0.6 * R + G/2.6 - B \qquad (3)$$

### 2.2.3. Regions of Interest

For the extraction and registration of the insects, the regions of interest (ROIs) property is applied, which allows such features as eccentricity, area, centroid, solidity, and others to be obtained [42,43].

## 3. Results

Table 1 shows the average temperature and number of pest type individuals detected during the experiment for each week. As shown in the table, a density of more than 1300 insects per week of a single type of pest can be detected, as well as a lower density of *Anthonomus eugenii*, and variable amounts of other types of pests. A relationship between temperature and the number of insects detected can also be observed.

**Table 1.** Weekly numbers of specimens by type of pest and average temperature.

| Time in Weeks | Temperature °C | Bactericera cockerelli | Thrips tabacci | Bemisia tabaci | Epitrix cucumeris | Anthonomus eugenii | Myzus persicae |
|---|---|---|---|---|---|---|---|
| | | | | Number of Specimens | | | |
| 1 | 21.87 | 12 | 56 | 36 | 1 | 0 | 7 |
| 2 | 23.69 | 13 | 58 | 38 | 1 | 1 | 9 |
| 3 | 21.27 | 11 | 54 | 34 | 1 | 0 | 5 |
| 4 | 17.91 | 94 | 455 | 48 | 8 | 2 | 36 |
| 5 | 17.4 | 203 | 983 | 62 | 12 | 1 | 71 |
| 6 | 17.06 | 174 | 865 | 83 | 12 | 2 | 70 |
| 7 | 17.83 | 114 | 565 | 150 | 3 | 1 | 68 |
| 8 | 17.59 | 100 | 553 | 130 | 3 | 1 | 60 |
| 9 | 19.83 | 109 | 565 | 140 | 4 | 2 | 64 |
| 10 | 19.83 | 107 | 560 | 140 | 3 | 1 | 63 |
| 11 | 18.59 | 92 | 1203 | 326 | 30 | 0 | 30 |
| 12 | 17.4 | 87 | 1309 | 357 | 33 | 0 | 23 |
| 13 | 19.87 | 57 | 826 | 261 | 29 | 2 | 16 |
| 14 | 19.41 | 18 | 181 | 128 | 15 | 1 | 11 |
| 15 | 18.73 | 18 | 178 | 128 | 16 | 1 | 10 |
| 16 | 17.9 | 18 | 178 | 128 | 16 | 1 | 11 |
| 17 | 17.67 | 18 | 175 | 128 | 16 | 1 | 10 |
| 18 | 18.03 | 18 | 175 | 128 | 16 | 1 | 11 |
| 19 | 17.18 | 11 | 100 | 72 | 9 | 0 | 6 |

Through the SIFT algorithm, tests were carried out to detect the different genera of insects (Table 2). In the images, five genera were found; in the data obtained in Week 1 of the experimentation, the percentages obtained in the detection reached an average precision of 90%.

**Table 2.** Detection percentages of specimens according to the five genera.

| Genus | Number of Insects Manual Method | Number of Insects Detected whit SIFT | Accuracy Rating |
|---|---|---|---|
| *Bactericera cockerelli* | 12 | 12 | 100% |
| *Thrips tabacci* | 56 | 28 | 50% |
| *Bemisia tabaci* | 36 | 36 | 100% |
| *Epitrix cucumeris* | 1 | 1 | 100% |
| *Myzus persicae* | 7 | 7 | 100% |

In the insect pest detection images, 17 insects of different species were found; the results (Table 3) for each genus were obtained, showing up to 100% accuracy for the detection of *Anthonomus eugenii*, *Myzus persicae*, and *Epitrix cucumeris*.

**Table 3.** Detection percentages of specimens according to the insect genus in an image with 17 insect individuals.

| Genus | Number of Insects Manual Method | Number of Insects Detected via SIFT | Accuracy Rating |
|---|---|---|---|
| *Bactericera cockerelli* | 3 | 2 | 67% |
| *Thrips tabacci* | 3 | 2 | 67% |
| *Bemisia tabaci* | 3 | 2 | 67% |
| *Epitrix cucumeris* | 3 | 3 | 100% |
| *Myzus persicae* | 3 | 3 | 100% |

Figure 2 shows the results after applying image processing, starting with the original image of each genus, CIE*L*A*B filter, and ROI, and applying the same processing to each type of pest detected.

**Figure 2.** Representation of each filter applied to each pest genus.

Table 4 shows the invariant characteristics of each genus, obtaining the mathematical values of each insect, and the size in square millimeters representing the approximate real size of each insect.

**Table 4.** Mathematical representation of other descriptors obtained by the genus of pests.

| Feature | | *Bactericera cockerelli* | *Thrips tabacci* | *Bemisia tabaci* | *Epitrix cucumeris* | *Anthonomus eugenii* | *Myzus persicae* |
|---|---|---|---|---|---|---|---|
| Solidity | | $8.60 \times 10^{-1}$ | $8.84 \times 10^{-1}$ | $8.09 \times 10^{-1}$ | $9.66 \times 10^{-1}$ | $7.59 \times 10^{-1}$ | $8.82 \times 10^{-1}$ |
| Eccentricity | | $8.78 \times 10^{-1}$ | $9.71 \times 10^{-1}$ | $7.22 \times 10^{-1}$ | $7.62 \times 10^{-1}$ | $7.99 \times 10^{-1}$ | $8.52 \times 10^{-1}$ |
| Pixel area | | $1.41 \times 10^{4}$ | $6.34 \times 10^{3}$ | $3.61 \times 10^{4}$ | $4.02 \times 10^{4}$ | $2.78 \times 10^{4}$ | $4.94 \times 10^{3}$ |
| Centroid | X | $1.02 \times 10^{2}$ | $1.18 \times 10^{2}$ | $1.33 \times 10^{2}$ | $1.03 \times 10^{2}$ | $1.62 \times 10^{2}$ | $1.11 \times 10^{2}$ |
| | Y | $1.30 \times 10^{2}$ | $1.96 \times 10^{2}$ | $2.26 \times 10^{2}$ | $1.30 \times 10^{2}$ | $1.96 \times 10^{2}$ | $1.38 \times 10^{2}$ |
| Perimeter | | $5.77 \times 10^{2}$ | $4.12 \times 10^{2}$ | $9.96 \times 10^{2}$ | $7.96 \times 10^{2}$ | $9.47 \times 10^{2}$ | $3.17 \times 10^{2}$ |
| Axis length | Major | $2.04 \times 10^{2}$ | $1.88 \times 10^{2}$ | $2.88 \times 10^{2}$ | $2.84 \times 10^{2}$ | $2.57 \times 10^{2}$ | $1.13 \times 10^{2}$ |
| | Minor | $9.75 \times 10^{1}$ | $4.54 \times 10^{1}$ | $1.99 \times 10^{2}$ | $1.84 \times 10^{2}$ | $1.54 \times 10^{2}$ | $5.94 \times 10^{1}$ |
| Size in mm$^2$ | | $4.78 \times 10^{-1}$ | $2.26 \times 10^{-1}$ | $9.12 \times 10^{-1}$ | 2.51 | $7.12 \times 10^{-1}$ | $1.03 \times 10^{-1}$ |

The solidity is a measurement of the overall concavity of an object, defined as the actual object area divided by the convex hull area. Thus, as the object becomes more solid, the image area and convex hull area approach each other, resulting in a solidity value of 1 [44–46].

The eccentricity is defined as the degree to which the object is similar to a circle, taking into consideration the smoothness of the perimeter. This might be denoted as the roundness because it has a clearly defined range of values. It is 0 for a perfectly round object and 1 for a line-shaped object [28].

The pixel area is the area of the insect measured in pixels.

The centroid X and Y is the point that is located in the center of a geometric body. In this case, the position of this point with respect to the *x*-axis and the *y*-axis is taken into account.

The perimeter is the length that surrounds an object; it refers to the sum of the pixels that limit its area in the case of objects detected in image processing.

The axis length represents the length of the major and minor axes of the object. The size in square millimeters is the size calculated with respect to the actual size of the insect and the area in pixels.

The values in the solidity column that belong to the genera *Thrips tabacci* and *Myzus persicae* are similar ($8.84 \times 10^{-1}$ and $8.82 \times 10^{-1}$, respectively), which means that under the same processing, the two types of insects have similar solidity; the same is the case for values in the centroid column corresponding to *Bactericera cockerelli* and *Epitrix cucumeris* ($1.02 \times 10^2$ in X and $1.30 \times 10^2$ in Y), but they can be differentiated by other characteristics.

These similarities are not of great importance since there are other descriptors (area, centroid, perimeter, etc.) with which confusion can be eliminated when making an automatic classification.

Figure 3 shows the correlation graphs between the data obtained via the manual method and those obtained via the method proposed in this investigation. It can be seen that the results of the two methods are similar.

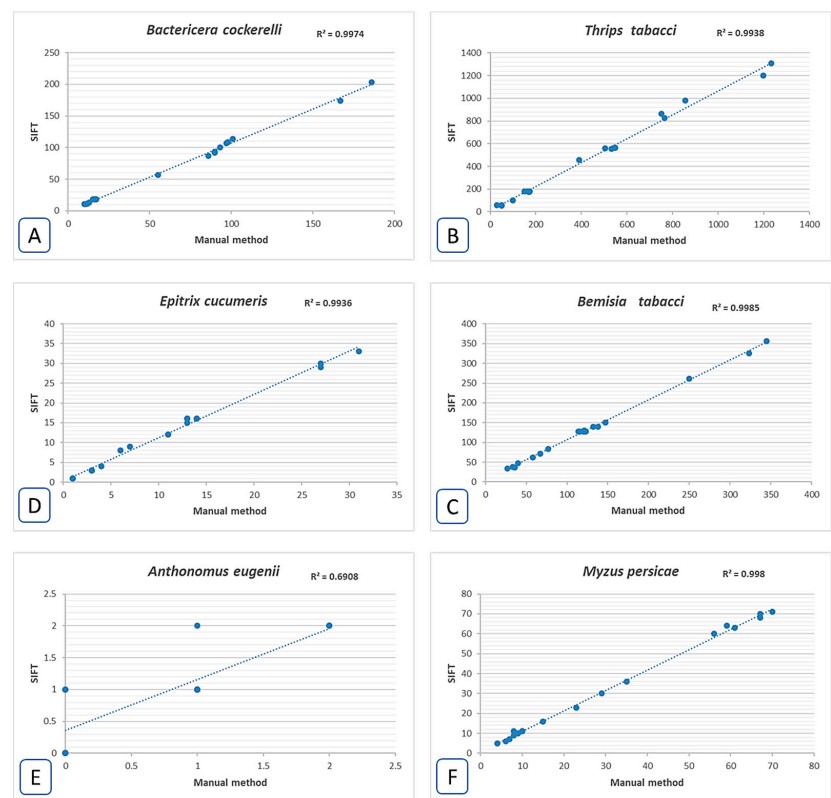

**Figure 3.** Correlation graphs between the manual method and SIFT.

## 4. Discussion

In various studies [6–11], the importance of detecting the presence of the pest insects present in mirasol chili crops is detailed and described. They are virus vectors and cause direct damage to the plant by sucking the sap; they also feed on the phloem of the plants, leaves and tender shoots. They are very common throughout the world and damage a great variety of crops of fruit plants, vegetables and ornamental plants. They are extremely dangerous for agriculture because they can cause losses of 10 to 100% of crop production. They cause defoliation, yellowing or reduction of vigor. During the winter, they lodge in weeds, under leaves or grass near the cultivated fields, and in spring, they feed on the new crops that are available; this is why weed clearing before and during the start of a new crop cycle and crop monitoring with various resources such as yellow sticky traps must be performed to prevent the presence of these insects.

This document presents a digital image processing method that involves two stages. In the first stage, the SIFT algorithm is applied to images of sticky yellow traps to extract the invariant characteristics of each insect contained in the trap, and to identify and differentiate the insect genera. For this stage, we showed that with the SIFT algorithm, different genera of insects can be identified with up to 94% detection of the specimens contained in the images.

In the second stage, image correction is applied to transform the original image to grayscale, then to the CIELAB color space, and finally to extract the invariant characteristics of each insect, as shown in Table 4.

Table 3 shows the mathematical representations of each feature according to the studied genus. For example, the solidity values of *Thrips tabacci* and *Myzus persicae* are similar ($8.84 \times 10^{-1}$ and $8.82 \times 10^{-1}$, respectively). However, without this, these two genera of insects can be differentiated by the other invariant features presented (eccentricity, the area in pixels, centroid, perimeter, size in square millimeters).

To analyze the efficiency of the proposed method, the correlations by type of pest detected are shown in Figure 3. In the case of *Anthonomus eugenii*, a very low correlation of $R^2 = 0.6908$ (Figure 3E) was observed in comparison with the other types of pests (Figure 3A–D,F), which may be due to the fact that the insect was found in smaller numbers during the experimentation period.

These data indicate that the digital image processing method presented in this document can be applied to specify the invariant characteristics of different common insects in mirasol pepper crops.

## 5. Conclusions

In this work, we used a digital image processing method to extract the physical characteristics, invariants, and descriptors specific to different pests that, according to our experiments in May to September 2018 (Table 1), are the most common among pepper crops in the state of Zacatecas.

The digital processing of sticky yellow trap images has been shown to help detect, classify and count different types of pest insects present in crops [11,19–21], so there is a need to apply this method in agricultural areas to help entomologists confirm their detections or classifications.

The SIFT algorithm detected the five genera in images of 17 insects of the species reported in this work with up to 94% accuracy, compared to the image with 5 insects of the species reported, in which an average of 100% accuracy was obtained, suggesting that more insects present in an image increases the accuracy of detection.

In Table 4, it can be seen that there were similarities in the characteristics between one type of pest and another. This could cause confusion but observing the rest of the characteristics will help to determine the type of pest present.

One of the most relevant results is the high coefficient of determination, from $R^2 = 0.69$ to $R^2 = 0.99$, among the different types of pests analyzed in this study. In the case of the

most deficient correlation (*Anthonomus eugenii*), this is due to the minimum number of specimens detected during the experimentation period.

The proposed method was tested and validated with images of traps containing the insects described in the results and by comparing with the data obtained by an entomologist.

One of the main drawbacks of the method is the digitization of the traps since they must be scanned under laboratory conditions to avoid contamination.

To obtain the approximate size of the insect, the digitization of the trap must always be performed with the same distance between the trap and the digitization medium since this characteristic is based on the number of pixels that a square centimeter of the trap occupies, and variation in the distance will affect the precision of this characteristic.

To try to solve these drawbacks, future work can be done on the structure for digitizing the trap, as well as on the design of the means of transport for the traps to avoid contamination.

The proposed method was tested and validated with images of traps containing the insects described in the results, but it is not limited to only these genera.

This method of digital image processing is expected to help entomologists to identify insects present in sticky yellow traps placed on various crops in a timely manner.

**Author Contributions:** M.M.-L. and C.L.C.-M., investigation; M.M.-L., methodology, data curation, and writing—original draft; G.E.-G., resources and supervision; C.A.O.-O. and H.A.G.-O., writing—review and editing; H.A.G.-O., visualization; L.F.L.-V., A.D.R.-D.S. and M.d.R.M.-B., resources; L.O.S.-S., conceptualization, project administration, data curation, and supervision; L.O.S.-S., agreed to be accountable for all aspects of the work in ensuring that questions related to the accuracy or integrity of any part of the work are appropriately investigated and resolved. All authors have read and agreed to the published version of the manuscript.

**Funding:** This research received no external funding.

**Institutional Review Board Statement:** Not applicable.

**Informed Consent Statement:** Not applicable.

**Acknowledgments:** The authors thank CONACYT for scholarship number 470427, and the National Laboratory CONACYT-SEDEAM. Special thanks to Salvador Rosas Gallegos for his valuable support in entomological aspects and teaching on the subject.

**Conflicts of Interest:** The authors declare no conflict of interest.

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
