# Peer review of "Extraction of Pest Insect Characteristics Present in a Mirasol Pepper (Capsicum annuum L.) Crop by Digital Image Processing"

_applsci, doi:10.3390/app112311166_

Round 1
Reviewer 1 Report
The study is very interesting and promising and method proposed in it has great potential and agricultural significance. But unfortunately the study is not very well written. Both the introduction and the discussion are extremely uninformative. The results section lacks clarity. Below are more specific comments. Introduction: To place the findings in context it would be useful to mention existing pest monitoring methods and explain how the new proposed method improves and augments the existing ones. Lines 51-56 This is superfluous, most readers will know how a scientific paper is organized Methods: It would be useful to describe how the insects were identified to confirm the accuracy of the image recognition process. Also, it would be useful to employ collection specimens for comparison to obtain realistic measurements. Results Lines 128-136 This paragraph is entirely unclear. Please rephrase. It’s really hard to understand how reliable or unreliable the proposed method it. Discussion Again, the findings should be placed in context, potential drawbacks and ways to solve them should be mentioned.Author Response
Response to Reviewer 1 Comments
Introduction:
Point 1:
To place the findings in context it would be useful to mention existing pest monitoring methods and explain how the new proposed method improves and augments the existing ones.
Response 1: Some previous methods on which this study is based were written
Point 2:
Lines 51-56 This is superfluous, most readers will know how a scientific paper is organized
Response 2: the paragraph was deleted
Point 3:
Methods:
It would be useful to describe how the insects were identified to confirm the accuracy of the image recognition process.
Response 3: the paragraph was rewritten and it is detailed how the insects of each trap were obtained
Point 4:
Also, it would be useful to employ collection specimens for comparison to obtain realistic measurements.
Response 4: Comparative graphs were integrated between the proposed method and the manual method
Point 5:
Results
Lines 128-136 This paragraph is entirely unclear. Please rephrase.
Response 5: the paragraph was rewritten
Point6:
It’s really hard to understand how reliable or unreliable the proposed method it.
Response 6: Sections were written to detail the proposed method
Point 7:
Discussion Again, the findings should be placed in context, potential drawbacks and ways to solve them should be mentioned.
Response 7: Possible drawbacks and ways to solve them are mentioned.
Reviewer 2 Report
I read the paper with great interest, and the system presented appears to produce the right results. However, there are some weaknesses in the way the results were presented and I feel that a little re-working of the information and some help with English wording will go a long way ( I am not a native English speaker so I can relate). Overall I feel it is a strong piece of work and it should be published. So please take my reviews with a grain of salt and know that the work is interesting and worth it. Also please consider adding more ecological notes to the discussion, some of the insect pests that you are able to detect with your system are really problematic for modern agriculture and being able to respond to low levels accurately to prevent explosive infestations is super valuable.

Author Response
Response to Reviewer 2 Comments
Introduction
Point 1:

The paper starts abruptly with a conflicting mixture of generalities and specific comments about weed cleaning.
Response 1: the paragraph was rewritten
Point 2:

Agriculture 4.0 – Unclear where this comes from…the whole first sentence it is too fuzzy for my taste. I think the second sentence would be a much stronger start.
Response 2: the paragraph was rewritten
Point 3:

Second paragraph: I do not understand why reduction of phytosanitary products use is linked directly to “weed cleaning” (which I also do not understand what it means). Do you mean weed sampling? Edge of the field sampling?
It seems you are trying to indicate that to know how much product to apply you need to know what is there in terms of pests…Am I correct?
Response 3: Regarding the third question, the answer is that you are correct, the paragraph was rewritten hoping to clarify the interrelation of agriculture 4.0, precision agriculture, the importance of knowing the pest present in the crop, knowing the appropriate phytosanitary and application of the right amount.
Point 4:

Then it seems you are concentrating on irrigated crops and greenhouse production, correct?
Response 4: The research focusing pepper plants are grown with irrigation, the paragraph has already been corrected
Point 5:

Line 47 Maybe change to “that affect the physiology and growth of Mirasol pepper and interfere with production of this crop.
Response 5: Line 47 has been changed to “that affect the physiology and growth of Mirasol pepper and interfere with production of this crop.”
Point 6:

Line 49 re-writing suggestion
This paper describes a digital image processing method for the extraction of invariant features of common insect pests that attack vegetable crops. The paper presents the specific descriptors utilized for each of the pest categories which allow us to detect, classify, and count insects in traps.
Response 6: Line 49 was re-written as suggested by the reviewer
Materials and Methods
Point 7:

Line 59 – Not sure what you mean by “a chili irrigation crop” – I presume is a field where chili are grown with irrigation. I also presume it is not a greenhouse, correct?
The placement of the cards was it 10 cm above the canopy of the plants?
Response 7: Line 59 has been changed to “For the detection of the different types and incidence of insect pests, yellow sticky traps placed 10 cm above the plant canopy ​of the pepper plants are grown with irrigation”
Point 8:

Line 67 and 72 seem to be repeated.
Response 8: Line 72 was eliminated and the paragraph was re-written
Point 9:

Line 76 Maybe it should be Transformation instead of Transform?
Response 9: The word transform cannot be changed by transformation because it is a technique established and published by author David G. Lowe in 1999, entitled "Object Recognition from local scale-invariant features" in the speech about the SIFT technique (Scale Invariant Feature Transform)
Point 10:

Line 77 You need to introduce SIFT a bit…Maybe start by saying: An algorithm was used to extract invariant features of the pest insects. In this study, we used the SIFT algorithm, described by XXXX in XXXX to examine features of the insects in different positions and orientations in the sticky cards. This procedure provided the information that was later used to detect objects and produce specific descriptors.
Response 10: The paragraph was enriched according to the reviewer's suggestions.
The paragraph the paragraph was rewritten as follows:
“SIFT is a method called Scale Invariant Feature Transform and described by David G. Lowe in 1999 [10] is used to extract invariant features in orientation and scale in images.
This approach transforms an image into a large number of local feature vectors, each of which is invariant to translation, scale or size, illumination, and image rotation. The feature vectors that are then used to detect objects and to obtain other descriptors such as signature or chain are also used and described by Moreno-Lucio et al., In 2019 and Solís-Sánchez et al., 2011 [11, 12]. The SIFT algorithm consists of five stages (Figure 1), described and explained by P. Flores and J. Braun in 2011 and D.G. Lowe (1999, 2004) [9, 10, 13, 14], these five stages are applied to an original image and to another image with which the characteristics.
We used the SIFT algorithm to examine features of the insects in different positions, size and orientations in the sticky cards. This procedure provided the information that was later used to detect objects and produce specific descriptors.
The steps of the SIFT algorithm are described below: ”
Point 11:

Not sure what exactly are you referring to by “scale” = absolute size of the insect or comparative size with respect to other insects caught in the traps?
Response 11: With "scale" we refer that regardless of the size of the insect in the image, the insect can be identified with the characteristics obtained
Point 12:

Line 86
Confused about the “teaching candidate” I suppose you mean computer learning?
Response 12: The phrase was removed
Point 13:

Results need some English formatting, but they are clearer than the introduction.
Response 13: The English language of the entire document was revised The document will be sent to the magazine's English edition service.
Point 14:

Some terms that will become very important in the discussion should be defined for the reader. One of these terms is “solidity”.
Response 14: The definition of solidity and eccentricity was written
Point 15:

Figure 2 seems like a clear place to expand on the filters and the strength of the SIFT and the results.
Response 15: A correlation table was added to compare the proposed method and the manual count
Point 16:

Table 3 also deserves more time and effort because it introduces the features and filters applied.
Response 16: Details added in section
Discussion
Point 17:

I think you can make more of the discussion than you did here. The fact that you had insects with similar solidity but can use other descriptors to identify them clearly and correctly is very important, but these results are presented too quickly and then the discussion appears weaker.
Response 17: The importance of detecting these pests was written
